

# Genome-scale metabolic reconstruction and metabolic versatility of an obligate methanotroph *Methylococcus capsulatus* str. Bath

Ankit Gupta[1,*], Ahmad Ahmad[2,3,*], Dipesh Chothwe[1], Midhun K. Madhu[1], Shireesh Srivastava[2] and Vineet K. Sharma[1]

[1] Department of Biological Sciences, Indian Institute of Science Education and Research Bhopal, Bhopal, India
[2] Systems Biology for Biofuels Group, International Centre For Genetic Engineering And Biotechnology, New Delhi, India
[3] Department of Biotechnology, Noida International University, Noida, India
* These authors contributed equally to this work.

Corresponding authors
Shireesh Srivastava,
shireesh@icgeb.res.in
Vineet K. Sharma,
vineetks@iiserb.ac.in

## ABSTRACT

The increase in greenhouse gases with high global warming potential such as methane is a matter of concern and requires multifaceted efforts to reduce its emission and increase its mitigation from the environment. Microbes such as methanotrophs can assist in methane mitigation. To understand the metabolic capabilities of methanotrophs, a complete genome-scale metabolic model (GSMM) of an obligate methanotroph, *Methylococcus capsulatus* str. Bath was reconstructed. The model contains 535 genes, 899 reactions and 865 metabolites and is named *i*MC535. The predictive potential of the model was validated using previously-reported experimental data. The model predicted the Entner–Duodoroff pathway to be essential for the growth of this bacterium, whereas the Embden–Meyerhof–Parnas pathway was found non-essential. The performance of the model was simulated on various carbon and nitrogen sources and found that *M. capsulatus* can grow on amino acids. The analysis of network topology of the model identified that six amino acids were in the top-ranked metabolic hubs. Using flux balance analysis, 29% of the metabolic genes were predicted to be essential, and 76 double knockout combinations involving 92 unique genes were predicted to be lethal. In conclusion, we have reconstructed a GSMM of a methanotroph *Methylococcus capsulatus* str. Bath. This is the first high quality GSMM of a Methylococcus strain which can serve as an important resource for further strain-specific models of the Methylococcus genus, as well as identifying the biotechnological potential of *M. capsulatus* Bath.

# INTRODUCTION

The problem of global warming needs urgent attention and requires a multifaceted approach including the control on the emission of greenhouse gases. Methane is among

the most potent greenhouse gases and has 21 times higher global warming potential than $CO_2$ over a 100 year period (*MacFarling Meure et al., 2006*). Atmospheric methane value has increased significantly from 722 ppb in the preindustrial era (in 1750) to 1,834 ppb in 2015 (*MacFarling Meure et al., 2006*). Efforts are underway to reduce methane emission into the atmosphere and to synthesize other useful chemical commodities from it. However, the conversion of natural methane to desired liquid products has been a technological challenge (*Clomburg, Crumbley & Gonzalez, 2017*; *Conrado & Gonzalez, 2014*). In this scenario, the use of "methanotrophs," microbes for methane mitigation, seems to be an appealing approach. Methanotrophs are organisms capable of surviving on methane as the sole source of carbon and energy. In addition, methanotrophs can also act as biocatalysts to convert methane to various value-added products such as high-protein feed and liquid biofuels. Furthermore, methanotrophs can also produce industrially important products and biopolymers such as polyhydroxybutarate, vitamins, carboxylic acids, single cell protein, and antibiotics using methane as the carbon source (*Fei et al., 2014*; *Strong, Xie & Clarke, 2015*). Despite these positives, bioconversion through methanotrophs faces multiple challenges of low energy and carbon efficiencies, and cultures with low productivity (*Conrado & Gonzalez, 2014*; *Shima et al., 2012*). However, with the advancements in synthetic biology, there is a potential to address these challenges as shown in some recent studies (*Bogorad, Lin & Liao, 2013*; *De la Torre et al., 2015*; *Shima et al., 2012*).

Methanotrophs perform the task of methane bioconversion by expressing membrane-associated (pMMO) or soluble (sMMO) forms of methane monooxygenase, which can activate methane by oxidizing it to methanol. The methanol is converted to formaldehyde, via the ribulose monophosphate (RuMP) in type I methanotrophs, or serine or Calvin, Benson, Bassham cycle (CBB) cycle in type II methanotrophs (*Haynes & Gonzalez, 2014*; *Shapiro, 2009*). *Methylococcus capsulatus* Bath (henceforth referred as Mcap), is a gram-negative type-I gammaproteobacterium with ability to grow optimally at higher temperatures of around 45 °C and even in very low oxygen concentrations (*Walters et al., 1999*). All these properties make this bacterium an optimal candidate to study methanotrophs, and to exploit its metabolic potential as cell factories for the synthesis of bioproducts.

A genome-scale metabolic model (GSMM) is a collection of most of the annotated metabolic reactions in form of a model which makes it possible to simulate cellular metabolic behavior under different conditions. The GSMM are reconstructed and analyzed by flux balance analysis (FBA) using tools such as Constraint-Based Reconstruction and Analysis (COBRA) Toolbox (*Becker et al., 2007*; *Schellenberger et al., 2011*). At present, only one validated GSMM for a methanotroph *Methylomicrobium buryatense* strain 5GB1 is available (*De la Torre et al., 2015*). For Mcap, the genome-scale biochemical network reconstructions based on automatic annotation pipelines are available in BioCyc and ModelSeed. However, the network information in these automated reconstructions is not curated and contains general reactions which are not specific to Mcap. More importantly, these automated models do not contain information on the reactions utilizing methane as the carbon source, which is one of the most significant properties of methanotrophs.

Therefore, these models are insufficient to explore the metabolic landscape of this organism. In this study, we present a detailed genome-scale metabolic reconstruction "*i*MC535" for the type I methanotroph *Methylococcus capsulatus* Bath, constructed using a systematic standard approach (*Ahmad et al., 2017*; *Shah et al., 2017*; *Thiele & Palsson, 2010*) and was validated using FBA.

## MATERIAL AND METHODS

### Model reconstruction

The metabolic reconstruction of *Methylococcus capsulatus* Bath (Mcap) was carried out using its annotated genome sequence obtained from NCBI. The complete reaction network was built using primary and review literature, biochemical databases such as KEGG (http://www.genome.jp/kegg/) and MetaCyc (http://metacyc.org/), and automated GSMM pipelines such as ModelSEED (*Henry et al., 2010*) and BioModels (*Chelliah, Laibe & Le Novere, 2013*). The model was manually constructed on a pathway-by-pathway basis, beginning from the core set of reactions (central metabolic pathways) and expanding it manually to include the annotated reactions and pathways associated with Mcap. Annotations for reaction IDs and compound IDs used in the constructed model were obtained from the KEGG database. The neutral formula for the compounds was obtained from KEGG database or EMBL-ChEBI database. The charge on the metabolites and charged formula were calculated at pH 7.2. All of the reactions included in the network were both elementally and charged balanced and were categorized into either reversible or irreversible. If available, the directionality for each of the reactions was determined from primary literature data, or from the KEGG database. The other reactions were assumed to be reversible. However, some reactions were made irreversible to resolve futile cycles in the model. The gene-protein-reaction assignments were made from the genome annotation and the KEGG database. Spontaneous reactions were included in the reconstruction if evidence suggested their presence, such as the presence of at least the substrate or product in the reconstruction, or on the basis of literature evidence. The final model included a list of metabolic reactions, required enzymes and their respective genes, gene protein reaction association, the major subsystem of the reaction, metabolite name, and compartment (p- periplasm or c- cytoplasm, e- extracellular). The model was converted into an SBML format recognizable by COBRA Toolbox for FBA.

### Biomass composition

Biomass components, their respective percentages in dry cell weight (DCW), references, and corresponding coefficients in biomass equation are summarized in Table S1. The relative percentages of deoxynucleotides were calculated using the GC content (63.58%) reported for *Methylococcus capsulatus* Bath (*Ward et al., 2004*). The amino acid composition in proteins, relative percentages of ribonucleotides, and ash were assumed to be similar to that in *Methylomicrobium buryatense* 5GB1 (*De la Torre et al., 2015*; *Gilman et al., 2015*). As literature references on the composition of carbohydrates, cell wall components (LPS and peptidoglycan), vitamins, and cofactors in DCW for *Methylococcus capsulatus* Bath are unavailable, these components were assumed to be in the same proportion as

reported for *Methylomicrobium buryatense* 5GB1 (*De la Torre et al., 2015*; *Gilman et al., 2015*; *Puri et al., 2015*), *Methylomicrobium alcaliphilum* 20Z (*But et al., 2013*; *Ivanova et al., 2006*; *Kalyuzhnaya et al., 2013*; *Khmelenina, Kalyuzhnaya & Trotsenko, 1997*), *Methylomonas methanica* (*Khmelenina et al., 1994*; *Trotsenko & Shishkina, 1990*), *Methylobacterium extorquens* AM1 (*Peyraud et al., 2011*) or *Escherichia coli* (*Wientjes, Woldringh & Nanninga, 1991*), in proportions as shown in the Table S1. The lipid content was assumed to be the same to that in *Methylomicrobium buryatense* 5GB1 (*De la Torre et al., 2015*) (Table S1). The fatty acid composition of the phospholipids was compiled from primary literature (*Makula, 1978*; *Trotsenko & Shishkina, 1990*). For phosphatidylcholine and cardiolipin, the building blocks (fatty acids) were assumed to be the same as phosphatidylglycerol and ethanolamine and each species was supposed to be in equal proportion. The fatty acid composition for other lipid species was considered to be similar to that of phospholipids. Calculations related to the biomass equation are provided in Table S1.

## Model simulation and analysis

### Flux balance analysis

A network can be converted into a mathematical form that can be read by software like COBRA toolbox. This mathematical representation is a matrix of order $m \times n$, where m is a number of metabolites (rows), $n$ is the number of reactions (columns), and each element as the coefficient of a metabolite in a reaction. A negative coefficient corresponds to consumption or intake of a metabolite, whereas a positive coefficient corresponds to its production. This stoichiometric matrix (S) is a linear transformation of the flux vector ($v = v_1, v_2, ..., v_n$) to the vector of rate of change of concentration vector (X) with respect to time $\frac{d}{dt}X = S \cdot v$.

Where, $X = [x_1, x_2, ..., x_m]$ is the vector of intracellular metabolite concentrations.

With the psuedo-steady state assumption, this differential equation changes to a linear equation of the form ($S \cdot v = 0$). Other than these constraints, flux vector for each reaction can be assigned a range that defines the maximum and minimum flux values it can take ($v_{i,min} \leq v_i \leq v_{i,max}$). These constraints were used to define irreversible reactions by setting both the $v_{i,\min}$ and $v_{i,\max}$ as non-negative. Similarly, transport reactions for nutrient uptake were set as negative and reactions for by-product secretion were set as positive.

### Flux variability analysis

A metabolic network can attain more than one flux distributions at optimal biomass flux. Minimum and maximum flux values for each reaction constituting network can be computed using flux variability analysis (FVA). In this work, we separately minimized and maximized flux through each reaction while keeping the growth rate constant at 0.37 h$^{-1}$ (maximum reported specific growth rate (*Joergensen & Degn, 1987*)) and the methane intake rate fixed at 28.10 mmol/gDCW/h.

### Gene essentiality analysis and knock-out simulations

In silico single and double gene knockout studies were performed on the model. For the single gene deletion studies, each reaction was deleted one at a time by constraining the flux bounds to zero, and the metabolic network was tested for growth using FBA.

A reaction was defined as essential if the mutant showed ($\leq 10^{-6}$) growth rate. Essential reactions for Mcap in different conditions like optimal, oxygen stress, different nitrogen sources, and under deleted Embden–Meyerhof–Parnas (EMP) cycle were determined using these deletion studies. In addition to a genome-scale single reaction deletion study, a genome-scale double-deletion study was also performed using the doubleGeneDeletion function of the COBRA Toolbox.

## Energy parameters and main energetic assumptions

Bacteria require energy in the form of ATP for growth (e.g., macromolecular syntheses such as DNA, RNA, and Protein) and cellular maintenance. Depending on the growth rate of cells, growth associated ATP maintenance could vary, hence in this study, two different conditions were tested (i) ATP maintenance value of 23 mmol/gDCW (same as the low value used for *E. coli* (*De la Torre et al., 2015*; *Varma, Boesch & Palsson, 1993*)), and was considered as low ATP maintenance, (ii) ATP maintenance value of 59.81 mmol/gDCW, and was considered as wild type network (same as high value used for *E. coli* (*Feist et al., 2007*)). To account for non-growth-associated maintenance requirement, an ATP hydrolysis reaction was added to the model and the flux of this reaction was fixed to 8.39 mmol/gDCW/h (*Feist et al., 2007*). The ATP is produced by the ATP Synthase complex which involves the translocation of protons between periplasm of cell membrane and cytosol.

## Evaluation on nitrogen and carbon sources

To evaluate the metabolic versatility of Mcap, different nitrogen and carbon sources were tested and the growth rate was observed. For testing different carbon sources, the methane uptake was set to zero and the flux for the respective carbon source was set to 28.10 mmol/gDCW/h (the methane uptake at the optimal growth of 0.37 h$^{-1}$) unless mentioned otherwise in the result section. Similarly, in the case of different nitrogen sources, the flux value for the uptake of the default nitrogen source, that is, nitrate, was set to zero and the lower bound of the reaction flux for the respective nitrogen source was set equal to the uptake rate of nitrate in optimal growth conditions (0.37 h$^{-1}$).

### Identification of Co-sets

Correlated sets (Co-sets) are the sets of reactions whose fluxes remains functionally correlated, that is, remain correlated in all possible states of the network under a condition. The Co-sets for Mcap model were calculated by the COBRA Toolbox function "identifyCorrelSets." We used uniform random sampling using the Artificial Centering Hit-and-Run sampler while keeping the maximum correlation ($R^2$) to 1.0 for the sampling to be used for "identifyCorrelSets."

## RESULTS

### Characteristics of the metabolic reconstruction

The metabolic reconstruction for *Methylococcus capsulatus* Bath (Mcap) "*i*MC535" was carried out based on its annotated genome sequence, available literature, information from biochemical databases such as KEGG and BRENDA, and automated GSMM pipelines

**Table 1 Properties of metabolic reconstruction of *i*MC535.**

| Property | # |
| --- | --- |
| Total reactions | 899 |
| Enzymatic reactions | 766 |
| Exchange reactions | 61 |
| Transport reactions | 72 |
| Number of genes | 535 |
| Gene associated metabolic reactions | 714 |
| Non-gene associated metabolic reactions | 46 |
| Number of metabolites | 865 |

such as ModelSEED and BioModels (see Methods section for more details). The final constructed model contained 899 reactions corresponding to 535 genes with 865 metabolites distributed over two different cellular compartments: cytoplasm and periplasm (Table 1). The extracellular space was used as a pseudo-compartment. Depending on the cellular localization, each metabolite was placed in one or more of the three compartments, and the flux across the outer and inner membranes was enabled by transport reactions. Out of the 899 reactions, 766 were associated with metabolic conversions, while 133 were required for the transport or exchange of nutrients. A total of 46 non-annotated metabolic reactions were included in the model because either they were known to occur in Mcap based on the literature reports, or were required to fill the gaps in the network (Table S2).

All reactions constituting the model were distributed into 14 systems on the basis of major metabolic properties of Mcap (Fig. 1). The majority of the reactions were associated with amino acid, exchange, carbohydrate, lipid, and nucleotide metabolism. The information for each of the pathways identified downstream of core metabolism was verified manually. Central metabolic pathways, including carbohydrate metabolism, energy metabolism, and biosynthetic pathways for fatty acids, amino acids, nucleic acids, lipids, cofactors, and vitamins along with other correlated pathways, were manually reconstructed. Similarly, the pathways related to nitrogen uptake, utilization of sulfate, ammonia, nitrate, urea, and phosphate were also manually reconstructed. Each reaction was tested for mass balance and irreversibility constraints. The complete functional model "*i*MC535" is provided in Table S3 in a Spreadsheet format, and in File S1 in SBML format.

## Metabolic versatility and energy source determination

Flux balance analysis is widely used to analyze the reconstructed metabolic networks. The model performance was evaluated by comparing in silico performance with previously reported experimental observations for growth and response to different carbon, nitrogen, and energy sources.

### Methane utilization

A summary of methane metabolism and central metabolic pathways is shown in Fig. 2. In Mcap, methane can be oxidized by both sMMO and pMMO depending on the copper concentration in the environment. It has been reported that the growth rates for Mcap lie

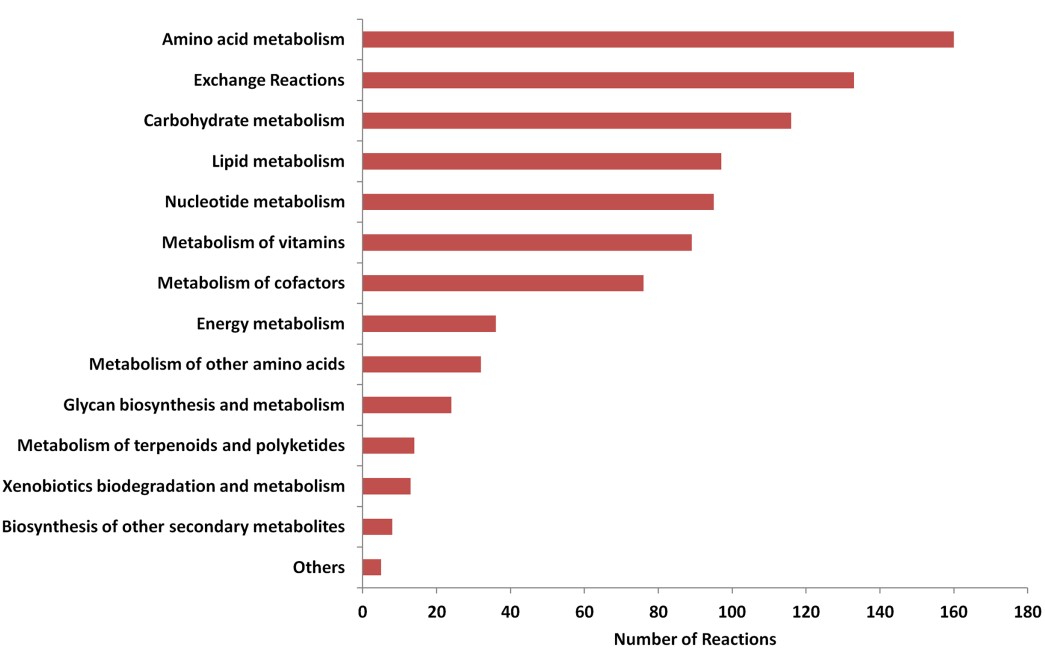

**Figure 1  Distribution of reactions in metabolic subsystems.**

between 0.25 h$^{-1}$ and 0.37 h$^{-1}$ (*Joergensen & Degn, 1987*). However, the experimental values of methane and oxygen consumption rates at the maximum-reported growth rate are not available. When we used the methane consumption rate of 18.5 mmol/gDCW/h, which has been reported for a related bacterium (*Methylomicrobium buryatense* 5GB1) (*De la Torre et al., 2015*), the growth rate was simulated to be 0.23 h$^{-1}$ (similar to Mb5GB1), and the oxygen uptake at this growth was 23.26 mmol/gDCW/h. The methane uptake rate of 28.10 mmol/gDCW/h was predicted at the optimal growth rate of 0.37 h$^{-1}$, and was used for all the downstream analysis. Therefore, the methane intake rates of Mcap are expected to be higher than that for *Methylomicrobium buryatense* 5GB1. As the FBA maximized the biomass yield on methane, in reality the biomass yield is expected to be lower. That is, the methane intake rate is very likely higher than that suggested by FBA. The model-calculated ratio of uptake values of oxygen and methane corroborates with the available experimental values which suggests that almost an equal or a slightly higher amount of oxygen is required as compared to methane for the optimum growth of Mcap (*Islam et al., 2015*).

The growth rates of 0.28 h$^{-1}$ and 0.37 h$^{-1}$ were observed in the case of pMMO knockout (low copper) and sMMO knockout (high copper) simulated conditions, respectively (Table 2). These values corroborate well with the experimentally observed growth range of 0.25–0.37 h$^{-1}$ (*Joergensen & Degn, 1987*) in these conditions. In terms of carbon conversion efficiency (CCE), a 34.14% increase is observed from low-copper to high-copper simulated conditions, which is very close to the previously reported experimental value of 38% increase (*Leak & Dalton, 1986*). Since, NADH is a limiting factor for the growth efficiency in Mcap (*Leak & Dalton, 1986*), the lower growth observed in low-copper conditions when sMMO are active can be attributed to the utilization of NADH by sMMO

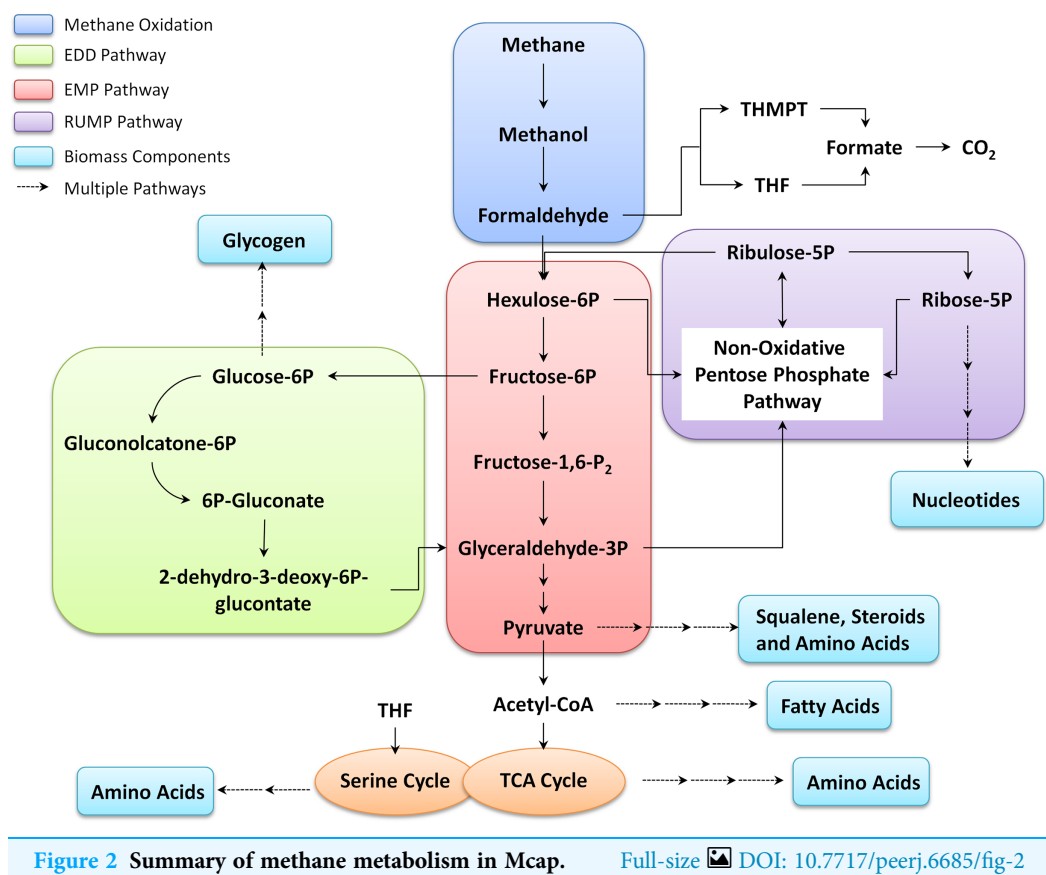

**Figure 2 Summary of methane metabolism in Mcap.**

for the oxidation of methane. On the contrary, in high-copper conditions, the higher growth efficiency is attributed to a reduced NADH requirement for methane oxidation, since pMMO uses quinones for the oxidation of methane (*Larsen & Karlsen, 2016*; *Shiemke et al., 1995*; *Zahn & DiSpirito, 1996*). Additionally, *i*MC535 also predicts the release of 0.37–0.39 moles of $CO_2$ per mole of methane consumed at the optimum growth range of 0.25–0.37 $h^{-1}$, which is in agreement with the previously-reported experimental value of more than 0.3 moles of $CO_2$ per mole of methane consumed (*Whittenbury, Phillips & Wilkinson, 1970*). These results further validate the reconstructed model.

After the oxidation of methane into methanol, the next step of formaldehyde formation is catalyzed by ferricytochrome c-dependent methanol dehydrogenase, which can be utilized by one of the following pathways: Tetrahydromethanopterin pathway (THMPT), Tetrahydrofolate (THF)-linked carbon transfer pathways, RuMP cycle, or Pentose phosphate pathway (PPP). In Mcap, formaldehyde is oxidized to $CO_2$ via either THMPT or THF pathway and is assimilated in biomass through the RuMP. The flux distribution shows that 56.2% of the formaldehyde goes to the RuMP pathway and 43.2% goes to the THMPT cycle. Thus, the RuMP pathway is 1.2 times more active than THMPT pathway (Fig. 3). Mcap has three variants for pathways downstream of the RuMP cycle: Entner–Doudoroff (ED) pathway, EMP pathway, and Bifidobacterial shunt (BS) (*Kao et al., 2004*). All these downstream pathways are interconnected through the transaldolase reactions of the PPP and result in ribulose-5-phosphate regeneration or pyruvate synthesis.

**Table 2 Computational predictions for different knockout simulations and growth conditions.**

| | Biomass flux (h$^{-1}$) | O$_2$ consumption (mmol/gDCW/h) | O$_2$:C ratio | CO$_2$ production | Nitrate consumption (mmol/gDCW/h) | CCE (%) |
|---|---|---|---|---|---|---|
| Wild type network | 0.37 | 34.18 | 1.22 | 12.54 | 3.41 | 55.36 |
| Low ATP maintenance | 0.44 | 29.85 | 1.06 | 9.49 | 4.07 | 66.23 |
| **Knockout simulations** | | | | | | |
| sMMO knockout | 0.37 | 34.18 | 1.22 | 12.54 | 3.41 | 55.36 |
| pMMO knockout | 0.28 | 39.77 | 1.42 | 16.49 | 2.54 | 41.30 |
| EMP knockout | 0.36 | 34.49 | 1.23 | 12.77 | 3.36 | 54.57 |
| **Carbon source** | | | | | | |
| Acetate | 0.59 | 21.12 | 0.38 | 31.42 | 5.42 | 44.09 |
| Methanol | 0.43 | 16.40 | 0.58 | 9.91 | 3.98 | 64.74 |
| Formaldehyde | 0.28 | 11.67 | 0.42 | 16.49 | 2.54 | 41.30 |

**Note:**
The intake flux for each of the carbon source was set as 28.10 mmol/gDCW/h.

We observed that the flux through PPP and ED pathway is almost three times more than that of glycolysis (Fig. 3). ED pathway is important as it supports the only known NADP-linked reaction in C-1 oxidation in Mcap via 6-phosphogluconate dehydrogenase, and thus could serve as the NADPH source for biosynthesis (Kao et al., 2004). The model did not show any growth in ED pathway knockout simulations, which is mainly because 6-phospho-D-gluconate (an intermediate of this pathway) is required for the growth of this bacterium, and can only be synthesized by the enzyme phosphogluconolactonase present in the ED pathway. Additionally, the gene for gluconokinase, which provides an alternate source for 6-phospho-D-gluconate, is also absent in the Mcap genome and hence, in the absence of any alternate source, the ED becomes essential for Mcap. On shutting down the EMP pathway, the predicted growth rate was only slightly reduced to 98.56% of the optimal growth (Table 2).

The ratio of the activity of the enzymes involved in the ED (6-Phosphogluconate dehydrase/phospho-2-keto-3-deoxygluconate aldolase) and EMP pathway (Phosphofructokinase and Fructose diphosphate aldolase) has been reported as 3.3:1 (Kao et al., 2004). With methane intake of 28.10 mmol/gDCW/h (at a growth rate of 0.37 h$^{-1}$), our model showed flux through ED and EMP pathways at a similar ratio of 3.4:1 without any external constraints, and hence corroborates very well with the available literature evidence. Additionally, in the robustness analysis, it was observed that the ED:EMP ratio can go up to a maximum of 10.9:1 for the optimal growth of 0.37 h$^{-1}$. All these observations are consistent with the biochemical studies of gammaproteobacteria, which point toward the role of ED pathway as a major route for C-1 assimilation even though the energy requirement of this pathway is greater (Anthony, 1982; Boden et al., 2011; Strom, Ferenci & Quayle, 1974). The flux through BS under normal simulation conditions was zero, suggesting that this pathway was non-essential in this organism.

An intriguing property which makes this bacterium unique is the presence of genes of both RuMP cycle and serine cycle. While the former plays a major role in biomass production, the role of the serine cycle is still unclear. It has also been hypothesized in

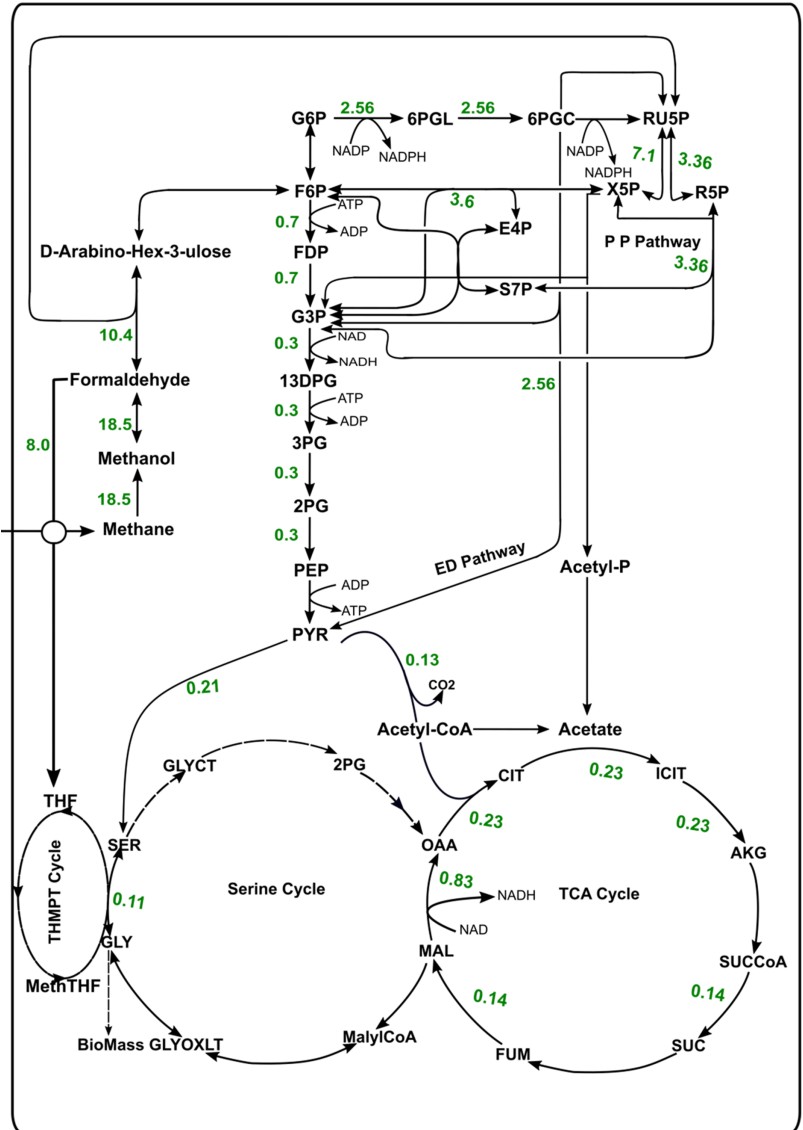

**Figure 3 Flux distribution through the central metabolic pathways of Mcap.**

earlier studies that serine cycle cannot operate as the major assimilatory pathway in Mcap unless glyoxylate, which is depleted during the cycle, is regenerated. However, no genes were identified in the Mcap genome which can convert acetyl-CoA to glyoxylate (*Chistoserdova, Vorholt & Lidstrom, 2005*). In our unconstrained network, we observed that all the reactions involved in the serine cycle had non-zero flux values and hence, were active. However, it is also important to note that all these reactions were also involved in other major pathways, and hence this observation points toward a secondary role of reactions of this cycle. In the case of serine cycle knockout simulations in *i*MC535, it was observed that only one reaction of this cycle was essential for the growth of Mcap, and had a role in glycerate synthesis and purine metabolism. These observations of flux distribution are consistent with the fact that *Methylococcus capsulatus* bath is type I

**Table 3 Computational predictions on different nitrogen sources and methane as a carbon source.**

| Nitrogen source | Biomass flux ($h^{-1}$) | $O_2$ consumption (mmol/gDCW/h) | Nitrogen consumption (mmol/gDCW/h) | CCE (%) |
|---|---|---|---|---|
| Alanine | 0.37 | 49.50 | 3.41 | 40.60 |
| Asparagine | 0.38 | 35.79 | 3.41 | 78.00 |
| Cysteine | 0.37 | 54.61 | 3.41 | 40.60 |
| Glutamine | 0.58 | 41.64 | 2.68 | 58.98 |
| Glutamate | 0.37 | 54.61 | 3.41 | 34.48 |
| Valine | 0.37 | 59.72 | 3.41 | 34.47 |
| Aspartate | 0.37 | 49.50 | 3.41 | 37.29 |
| Nitrate | 0.36 | 32.95 | 3.34 | 54.25 |
| Nitrite | 0.37 | 34.18 | 3.41 | 55.36 |
| Ammonia | 0.44 | 36.21 | 4.02 | 65.43 |

methanotroph and a member of the gammaproteobacteria family which utilizes RuMP pathway instead of the serine pathway for formaldehyde assimilation.

### Alternate energy and nitrogen sources

We simulated the growth of Mcap on different carbon and nitrogen sources to determine its metabolic versatility. The different carbon sources, such as methanol, formaldehyde, formate, acetate, and $CO_2$, were tested on iMC535. The model showed growth on reported downstream oxidation products of methane such as methanol and formaldehyde (*Eccleston & Kelly, 1972*), which could be assimilated in biomass via the RuMP pathway (Table 2). Further, acetate could also be utilized for growth, which is in accordance with a previously-published report (*Eccleston & Kelly, 1972*). The model showed no growth on $CO_2$.

The maintenance ATP (mATP) values are not known for Mcap. Therefore, we tested the response of the model to different mATP values. At low growth-associated maintenance ATP demand (GAM) value of 23 mmol/gDCW, the growth rate and the CCE were higher while reduced $O_2$ consumption and $CO_2$ production were observed compared to the wild type network (Table 2).

Similarly, iMC535 was tested on different nitrogen sources such as nitrate, nitrite, and ammonia. It was observed that using ammonia as the nitrogen source, the flux through biomass was higher as compared to nitrate/nitrite, and the CCE of Mcap was also increased (Table 3). The reason for these observations could be that extra NADH is required for nitrate assimilation as compared to ammonia. Another property, which makes Mcap unique as compared to the other known methanotrophs, is its ability to utilize amino acids such as Alanine, Asparagine, Cysteine, Glutamine, Glutamate, Aspartate, and Valine, as a nitrogen source for its growth (*Patel & Hoare, 1971*). We tested our model for the growth on each of these reported amino acids as the nitrogen source, and methane as the carbon source. The lower bound for amino acid intake was set as equal to the intake of nitrate (3.4 mmol/gDCW/h) at optimal growth conditions when nitrate was used as a nitrogen source. Growth was predicted in all the cases as shown in Table 3 since all these amino acids could also form pyruvate (*Eccleston & Kelly, 1972*; *Patel & Hoare, 1971*),
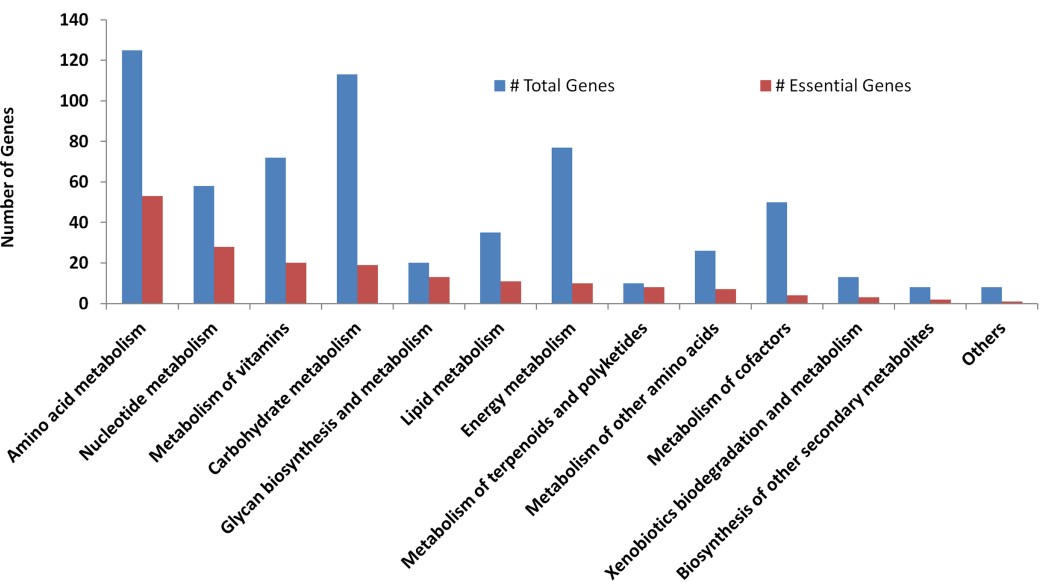

**Figure 4 Distribution of total genes and essential genes in metabolic subsystems.**

which is a nodal point in the metabolic network for the production of all the necessary biomass components. Among all the amino acids tested, the maximum growth of 0.58 h$^{-1}$ was observed when glutamine was utilized as the nitrogen source (Table 3). However, the model could not grow on other nutrient sources, such as glucose and fructose, since their transporters were absent in the genome, and hence were not included in the model.

## Analysis of essential genes and flux variability in Mcap

*i*MC535 was used as a framework to identify the candidate essential genes in Mcap. To simulate deletion of the reactions associated with each gene, the flux through the reaction was set to zero and the biomass reaction was maximized by FBA. Out of the 535 metabolic genes, the deletion of 161 (30.1%) genes was predicted to be lethal (biomass flux less than 10$^{-6}$) (Table S4; Fig. 4). As expected, gene deletions in central metabolic processes had a highly negative impact. However, most of the single gene deletions for methane oxidation were not lethal, since multiple copies of these genes were present in the genome. Interestingly, nitrite reductase gene was found essential for Mcap when it was grown on nitrogen sources such as nitrite and nitrate, whereas this gene was not essential when the model was grown on amino acids as the nitrogen source. The majority of the essential genes belonged to pathways associated with amino acid metabolism, carbohydrate metabolism, nucleotide metabolism, and metabolism of vitamins and cofactors. In addition to single-gene deletion study, double gene deletion study was also carried out. A total of 103 unique gene pairs showed multiple lethal/sick interactions, that is, growth rate less than 0.01 h$^{-1}$ with respect to either of the single gene deletion condition, out of which 76 interactions were lethal (Table S5). The genes showing lethal interactions were mostly involved in amino acid metabolism, exchange, carbohydrate metabolism, lipid metabolism, nucleotide metabolism, and metabolism of vitamins (Table S5).

In a metabolic network, optimal flux through objective function can be achieved through a number of alternative flux distributions if the alternate routes for the same function are available in the bacterium. Such variability in the solution space can be explored using FVA, which computes the range of allowed flux values for each reaction while keeping the optimum value of growth rate. This analysis revealed that 99 out of 383 active enzymatic reactions could show more than 0.1 mmol/gDCW/h flux change without affecting the growth rate. Other 15 enzymatic reactions showed variability between $10^{-4}$ and $10^{-1}$ mmol/gDCW/h. A total of 158 enzymatic reactions with zero flux could potentially carry flux in an alternate optimal solution space. On the other hand, 269 reactions had fixed flux (less than $10^{-4}$ mmol/gDCW/h variation) at optimal growth value and represented 70.23% of the active network. High variability was observed for reactions related to glucose, fructose, respective phosphates and bisphosphates because alpha-D, beta-D, and D compounds of above-mentioned species were considered as different entities. Glycolysis, EMP, EDD, serine cycle, fatty acid synthesis, etc., showed high variability due to alternate reactions or presence of the above entities.

## Network topology and reaction subsets

In the metabolic network, only a few metabolites were found to be highly connected, whereas the majority of the metabolites showed limited connections. The highly connected metabolites were referred to as hubs, which are crucial to the network since they connect the other metabolites (with few connections) to the central metabolism. Top 35 highly connected metabolites out of 827 unique metabolites in the *i*MC535 model are shown in (Fig. S1). Ranking of most of the highly connected metabolites in Mcap is similar to that observed in other models (*De la Torre et al., 2015*). As apparent, Mcap model also had the common metabolic hubs such as water, energy carrying molecules (ATP, NADH, NADPH, etc.), CoA, pyruvate, etc., which were expected due to their common functions in the metabolic network. Interestingly, amino acids, such as glutamate, glutamine, aspartate, alanine, and serine also showed high connectivity (*Patel & Hoare, 1971*).

In a metabolic model, reactions with linearly correlated fluxes form the correlated reaction set (Co-set) (*Marashi & Hosseini, 2015*; *Price, Schellenberger & Palsson, 2004*). Earlier studies on reaction correlation showed that uniformly sampled fluxes can be used to explore the regulatory properties of reactions in metabolic models (*Barrett, Price & Palsson, 2006*; *Barrett, Herrgard & Palsson, 2009*). There are 88 Co-sets present in the Mcap model, out of which the largest Co-set had 146 correlated reactions. Further analysis of this Co-set revealed that these reactions were involved in the synthesis of biomass precursors, and hence showed a correlation in all the conditions considered for this analysis. There were 17 reactions present in Co-set 2 participating in fatty acid biosynthesis, 12 reactions in Co-set 3 participating in lipopolysaccharide biosynthesis, six reactions in Co-set 4 participating in methane metabolism and five reactions in Co-set 5 participating in sulfur metabolism. From Co-set 6 onward, there were 7, 14, and 62 Co-sets having four, three and two reactions, respectively. The summary and detailed information of Co-sets and its participating reactions is given in Table S6.

## DISCUSSION

The model analysis revealed several interesting features about this organism. $i$MC535 could not simulate growth in the absence of ED pathway. It is because, 6-phospho-D-gluconate, an intermediate of ED pathway, is crucial for the growth of Mcap, and the gene (gluconokinase) for its alternate source was absent in the Mcap genome. Similarly, the analysis identified that $NH_3$ could be a better N-source than $NO_{3-}$ and that Mcap could grow on amino acids.

The properties of $i$MC535 model were also compared with $i$Mb5GB1 (*Methylomicrobium buryatense* strain 5GB1) model, which is the only available model for a methanotroph. The iMb5GB1 network is available in two formats: a network of 841 reactions that cannot be simulated, and a smaller COBRA toolbox-compatible model containing 402 reactions (http://sci.sdsu.edu/kalyuzhlab/computational-models/). Hence, the latter was used for this analysis. The number of reactions in the Mcap model (898 reactions) is more than twice of the working (that can be simulated) 5GB1 model (402 reactions). The models were compared using pathways and the number of reactions in 12 different subsystems (Fig. S2). The reactions that do not belong to these 12 categories were categorized as "Others." As expected, most of the subsystems in $i$MC535 had about two-fold or more the number of reactions compared to $i$Mb5GB1. For example, there are 32 reactions belonging to cysteine and methionine pathway in Mcap model, while only five reactions in the 5GB1 model. The subsystems such as nucleotide metabolism, vitamin metabolism and the "Others" had over three-times the number of reactions in our model compared to the $i$Mb5GB1 model. This additional information becomes important while identifying the essential genes as well as targets for metabolic engineering. For example, while 56.36% (177 out of 314) genes were found to be essential in the 5GB1 model, only 30% (161 out of the 535) genes were found essential in the Mcap model.

Bacteria can use several different molecules as electron acceptors and donors. For example, NADH, succinate and glycerol-3-phosphate can act as electron donors while quinone, ubiquinone, menaquinone, and nitrite can act as electron acceptors. This list is not exhaustive as other molecules may act as electron donors or acceptors. The gene for glycerol-3-phosphate dehydrogenase (menaquinone 8) was not identified in the genome sequence of *M. capsulatus* str. Bath according to KEGG database and therefore, this reaction was not included in the model. In our model ubiquinone-8 has been used as the general electron acceptor as has been done in some prior GSMMs (*Ahmad et al., 2017*; *Boyle & Morgan, 2009*; *Zuniga et al., 2016*), in addition to quinone and nitrite, while NADH and succinate have been used as electron donors. In our model, the ATP synthase complex reaction involves cytochrome c oxidase (ubiquinol) (RID: Cytochrome_c_oxidase), NADH dehydrogenase (ubiquinone) (RID: NADH_DH_ubi) and the ATP synthase (RID: ATPSynth) reactions. In the future, with the availability of more biochemical information, it would be possible to easily incorporate more reactions corresponding to electron donors and acceptors in the Mcap model.

Although the model developed in this study utilizes some parameters from the available published metabolic models, which might add some bias in the results. However, it is not

uncommon for models to use information from closely-related organisms, if such information is not available for the target organism, and nonetheless makes it possible to conduct some initial qualitative analyses at the genome scale. For example, the metabolic model of Tomato (*Solanum lycopersicum L.*) uses Non-growth associated ATP maintenance value of *Arabidopsis* model, and metabolic model of *Thermus thermophilius* uses growth-associated maintenance ATP demand (GAM) value of *E. coli*. (*Lee et al., 2014*; *Yuan et al., 2016*). The availability of more experimental data on Mcap, particularly on methane uptake rate and specific growth maintenance values, will help to improve the accuracy of *i*MC535 to predict cellular phenotypes. We anticipate that *i*MC535 model will serve as a framework for improved metabolic reconstructions, and as an analysis platform for the engineering of methanotrophs for methane mitigation and industrial applications.

## CONCLUSION

*Methylococcus capsulatus* is one of the two important experimental model systems to understand the biochemistry of methanotrophy. Thus, the first genome-scale metabolic reconstruction of *Methylococcus capsulatus* str. Bath, and constraint-based analysis of its metabolic network are the major steps in understanding the potential capabilities of these obligate methanotrophs and understanding methanotrophy in general. Improvements in the carbon assimilation pathways and channeling the metabolic flux of these pathways will be desired to fine tune its metabolic capabilities to various products of interest. Hence, the GSMM constructed in this study would play a pivotal role in designing and accelerating the metabolic engineering approaches to harness the potential of this methanotroph.

## ACKNOWLEDGEMENTS

The views expressed in this manuscript are that of the authors alone and no approval of the same, explicit or implicit, by MHRD should be assumed.

### Funding

This work was supported by intramural funding received from MHRD, Govt of India, funded Centre for Research on Environment and Sustainable Technologies (CREST) at IISER Bhopal. Ankit Gupta is a recipient of DST-INSPIRE Fellowship. The funders had no role in study design, data collection and analysis, decision to publish, or preparation of the manuscript.

### Grant Disclosures

The following grant information was disclosed by the authors:
MHRD, Govt of India, funded Centre for Research on Environment and Sustainable Technologies (CREST) at IISER Bhopal.
Ankit Gupta is a recipient of DST-INSPIRE Fellowship.

### Competing Interests

The authors declare that they have no competing interests.
## Author Contributions

- Ankit Gupta conceived and designed the experiments, performed the experiments, analyzed the data, contributed reagents/materials/analysis tools, prepared figures and/or tables, authored or reviewed drafts of the paper, approved the final draft.
- Ahmad Ahmad performed the experiments, analyzed the data, contributed reagents/materials/analysis tools, prepared figures and/or tables, authored or reviewed drafts of the paper, approved the final draft.
- Dipesh Chothwe performed the experiments, analyzed the data, contributed reagents/materials/analysis tools, prepared figures and/or tables, authored or reviewed drafts of the paper, approved the final draft.
- Midhun K. Madhu performed the experiments, approved the final draft.
- Shireesh Srivastava conceived and designed the experiments, authored or reviewed drafts of the paper, approved the final draft.
- Vineet K. Sharma conceived and designed the experiments, authored or reviewed drafts of the paper, approved the final draft.

## Data Availability

Raw data and code are available as Supplemental Files.

## Supplemental Information

Supplemental information for this article can be found online at http://dx.doi.org/10.7717/peerj.6685#supplemental-information.

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
