# Peer review of "Genome-scale metabolic reconstruction and metabolic versatility of an obligate methanotroph Methylococcus capsulatus str. Bath"

_PeerJ, doi:10.7717/peerj.6685_

## Round 0.1 · original submission · Major Revisions

Both reviewers find major problems with design of the model, from using non-existing reactions to lack of experimental support for parameters used. These need to be addressed before resubmission.

·

Basic reporting

The quality of language is satisfactory. The text of the manuscript is easy to read and understand. The paper is well organized, the literature references look sufficient.
Nonetheless, legends for all the figures and supplementary materials are absent. Please, add the legends or modify the figures so that they became more clear.

Also, there are some minor comments to the text.

1. Lines 72: It is better to write "the ribulose monophosphate (RuMP)" than "the ribulose monophosphate RuMP".

2. Lines 130-131: "relative percentages of ribonucleotides, and ash were assumed to similar to that in M. buryatense 5GB1". What is an "ash"?

Experimental design

Unfortunately, experimental design is so limited and biased, so this paper cannot be accepted without major revisions.

1. Lines 193-195. It is absolutely unacceptable to use such a reaction for the ATP production. This reaction can be used for the mitochondria but not for any bacterium. At first, Bacteria can use various electron acceptor, not only an oxygen (see e.g. Welte & Deppenmeier 2014, PMID: 24333786). Moreover, in lines 381-384 a nitrite reductase is mentioned. This points to oxygen is not an only electron acceptor that can be used by Methylococcus capsulatus str. Bath. Second, various reactions are possible even for oxygen-dependent respiration, e.g. compare https://www.vmh.life/#reaction/CYTBD2 and https://www.vmh.life/#reaction/CYOmq. Third, NADH is not an only possible electron donor. Thus multiple intermediates of the central carbon catabolism as succinate (https://www.vmh.life/#reaction/SUCDimq) or glycerol-3-phosphate (https://www.vmh.life/#reaction/G3PD6) are used as electron donors. Thus, usage of only one reaction for NADH oxidation, oxygen reduction, and ATP synthesis is absolutely unacceptable. Please, use separate reactions for an oxidation of electron donors, reduction of electron acceptors, and APT synthesis (e.g. https://www.vmh.life/#reaction/ATPS3).

2. Lines 344-345: "The different carbon sources such as methanol, formaldehyde, formate, acetate, and CO2, were tested on iMC535." - please, explain a selection of these carbon sources.

Validity of the findings

Because of ill-conditioned experimental design, it is not possible to estimate the validity of the findings.

Additional comments

No comments.

Reviewer 2 ·

Basic reporting

no comment

Experimental design

everything is fine

Validity of the findings

The MS has one serious issue preventing it to be published in its current state: the lack of relevant experimental data. Authors collected the experimental data (among highly important are ATP maintenance, methane consumption an other) for their model evaluation from i) published Methylococcus capsulatus str. Bath, ii) data, published for another methanotroph M. buryatense 5GB1 and iii) even from E. coli. All of these can add serious bias to current calculations making the model of low value. I understand that it is most probably impossible to obtain this information ASAP, however, authors can possibly obtain more available experimental data to evaluate it with the model.

Additional comments

There also several specific comments:

L 39. Why Methylococcus capsulatus str. Bath is unique?
L 53-54 It is still questionable
L 76 In microbiology 45C cannot be regarded as high temperature – it is closer to mesophily than thermophily
L 77 Again, uniqueness of M. capsulatus. Could you please list the unique features of this microorganism in comparison with other Methylococcaceae and other methanotrophs?
L 79 Not speaking of other methylotrophs and methanotrops in particular, the only Methylococcaceae family has already around 60 sequenced genomes in IMG or NCBI databases
L129-130 amino acid composition in proteins. Why not calculated in silico?
L177 what does it mean? 10-6 of what?
L 196 how this equation was calculated? from the calculations down the text?
L 236 what are these categories? look like COGs, but COGs weren`t mentioned in the methods
L 242-244 what does it mean? Reconstructed de novo? I assume authors manually verified the KEGG maps-based (or MetaCyc) automatic reconstructions.
L 440-441 It is strange since according to the original publication of Foster and Davis, 1966, NO3- was much better than ammonium or amino acids

---

## Round 0.2 · Minor Revisions

Overall the reviewers are satisfied with your revision. However, I'd suggest a more detailed discussion in the text of their concerns, as they are likely to cause similar questions from the readers. In particular, it might be useful to discuss in some detail the ATP-production reactions and provide references for the following statement: "Although the model developed in this study utilizes some parameters from the available published metabolic models, which might add some bias in the results. However, it is not uncommon for models to use information from closely-related organisms, if such an information is not available for the target organism...:.

Reviewer 2 ·

Basic reporting

everything is fine

Experimental design

everything is fine

Validity of the findings

I am not satisfied with the response on my main concern on relationship of the model and experimental data.

Additional comments

The answers on the particular questions are fine. Overall, the MS is well-written, designed etc. However, besides the main concern on the validity of the model, authors still did not convince me their study is of particular scientific novelty or impact.

---

## Round 0.3 · Minor Revisions

The revised text contains grammatical errors and misprints, e.g.:
"In future, as more biochemical information about the other electron donors and acceptors in M. cap becomes available".

The rebuttal letter cites the following sentence: "Metabolic model of Tomato (Solanum lycopersicum L.) use Non growth associated ATP maintenance (NGAM) value Arabidopsis model and Thermus thermophilius model uses growth associated ATP maintenance (GAM ) value of E.coli. [1,2]" - I could not find it in the paper; besides, the style of references in this sentence does not conform to the style elsewhere in the manuscript.

---

## Round 0.4 · accepted · Accept

All comments have been addressed.

#